# *Saccharomyces cerevisiae* as a Tool to Investigate Plant Potassium and Sodium Transporters

**DOI:** 10.3390/ijms20092133

**Published:** 2019-04-30

**Authors:** Antonella Locascio, Nuria Andrés-Colás, José Miguel Mulet, Lynne Yenush

**Affiliations:** Instituto de Biología Molecular y Celular de Plantas, Universitat Politècnica de València-Consejo Superior de Investigaciones Científicas, 46022 Valencia, Spain; antonella.locascii@gmail.com (A.L.); nuanco@btc.upv.es (N.A.-C.); jmmulet@ibmcp.upv.es (J.M.M.)

**Keywords:** potassium transport, sodium transport, plant ion channels, yeast, functional complementation, protein-protein interaction, heterologous expression

## Abstract

Sodium and potassium are two alkali cations abundant in the biosphere. Potassium is essential for plants and its concentration must be maintained at approximately 150 mM in the plant cell cytoplasm including under circumstances where its concentration is much lower in soil. On the other hand, sodium must be extruded from the plant or accumulated either in the vacuole or in specific plant structures. Maintaining a high intracellular K^+^/Na^+^ ratio under adverse environmental conditions or in the presence of salt is essential to maintain cellular homeostasis and to avoid toxicity. The baker’s yeast, *Saccharomyces cerevisiae*, has been used to identify and characterize participants in potassium and sodium homeostasis in plants for many years. Its utility resides in the fact that the electric gradient across the membrane and the vacuoles is similar to plants. Most plant proteins can be expressed in yeast and are functional in this unicellular model system, which allows for productive structure-function studies for ion transporting proteins. Moreover, yeast can also be used as a high-throughput platform for the identification of genes that confer stress tolerance and for the study of protein–protein interactions. In this review, we summarize advances regarding potassium and sodium transport that have been discovered using the yeast model system, the state-of-the-art of the available techniques and the future directions and opportunities in this field.

## 1. Introduction

Potassium (K^+^) and sodium (Na^+^) are two nutrients essential for plant life. They are absorbed from the soil through the roots, translocated to the rest of plant organs through movement into the xylem/phloem, to be finally compartmentalized in the organs where they exert their specific functions.

In particular, K^+^ is a macronutrient involved in fundamental processes required for plant growth and development. It is required for the activity of numerous enzymes involved in the generation of adenosine triphosphate (ATP) and for cellular turgor maintenance [1,2]. It also participates in the regulation of the opening and closing of stomata [3,4], facilitates protein and starch synthesis and is involved in the neutralization of inorganic and organic anions and macromolecules, thus maintaining pH homeostasis and controlling the electrical potential across the plasma membrane [5].

The respiration rate of plants constitutes the determining factor for proper K^+^ uptake and transport by plants [6,7]. In fact, it has been observed that the ion concentration in the xylem vessels, which is directly determined by the transpiration rate, limits the rate of transfer of K^+^ from the root cells to the xylem vessels [8]. It has also been observed that low K^+^ concentrations in the soil during a long growth period will produce plants with a higher transpiration rate and increased stomata frequency per leaf area; while, higher internal K^+^ is correlated with a lower transpiration rate and fewer stomata per leaf area [6].

Na^+^ is considered to be a micronutrient since it is not necessary in high amounts for plant growth. It is not required for C3 plants, but is essential for C4 species, where it participates in the carbon cycle, chlorophyll synthesis and photosystem II activity [9]. Nevertheless, Na^+^ ions become important in small amounts for C3 species when exogenous K^+^ is present at low concentrations in the soil. It has been shown that Na^+^ can replace K^+^-specific functions due to the fact that in their hydrated forms, Na^+^ and K^+^ are chemically and structurally very similar [10,11]. On the other hand, K^+^ homeostasis is severely affected by salt stress and high salt concentrations are toxic for plants. For this reason, plants have developed specific mechanisms to sense, transport and store both K^+^ and Na^+^ [12]. 

In order to avoid and/or limit Na^+^ toxicity, plants have evolved several uptake systems consisting of multiple channel types able to discriminate and coordinate the influx of these two different cations. For instance, it has been shown that Na^+^-permeable channels present certain flexibility in their Na^+^:K^+^ selectivity depending on the external concentration of salt, cooperating with other inward cation channels at the plasma membrane [10]. Under saline conditions, reducing the activity of these non-selective channels is thought to increase salt tolerance by impeding Na^+^ accumulation. 

In addition, plants have evolved uptake mechanisms able to adapt to large variations in the external concentrations of K^+^. In 1973, Epstein and collaborators proposed the presence of two mechanisms of K^+^ uptake that work simultaneously [1]. The first is a high-affinity system, mediated by carriers that are active when K^+^ concentrations are low (micromolar range), which was subsequently shown to operate through the coupled movement of K^+^ with other cations, such as H^+^ or Na^+^ [13,14]. The second is a low-affinity system that functions as a “passive” transporter, working when K^+^ concentrations are relatively high (millimolar range). 

Since then, many other studies have been published describing the molecular details regarding different K^+^ channels and transporters, which have been cloned and characterized in several plant species [15,16,17,18,19,20,21]. Most of our current knowledge on the physiological impact of K^+^ channel activities derives from studies carried out in model plants, such as *Arabidopsis thaliana*. Other organisms have been successfully used as tools for the functional characterization of these ion transport proteins, such as *Xenopus* oocytes and the baker’s yeast, *Saccharomyces cerevisiae*. 

The power of the budding yeast, *S. cerevisiae* in these kinds of studies resides not only in the multiple properties that make it an efficient, easy and attractive system in which to screen and select genes conferring certain phenotypes, but also in the fact that the majority of the subcellular processes governing cellular ion homeostasis in yeast cells are largely conserved in higher eukaryotes. Thus, insights from yeast can be easily translated to other organisms. In addition, *S. cerevisiae* allows for large-scale, genome-wide analyses in a fast and economically efficient manner. Work with *S. cerevisiae* allows for the discovery and/or characterization of many aspects of ion transporter function and regulation, but obviously the final physiological proof of yeast-based hypotheses need to be validated *in planta*. However, yeast remains an incredibly productive system to discover and study a multitude of aspects related plant ion transporter function and provides a platform for the generation of knowledge that can serve as the basis for experimental designs in more complex and time-consuming plant model systems.

For details on the studies of the physiological characterization of plant K^+^ channels both in vivo and in the *Xenopus* oocyte model, we refer the reader to other thorough reviews [22,23,24,25]. In this review, we will describe and summarize results obtained using four general experimental approaches employing *S. cerevisiae* that have been successfully applied to identify and/or characterize plant K^+^ and Na^+^ transport proteins and their regulators: Functional complementation using mutants, high-throughput protein­–protein interaction assays, reconstitution of functional transport systems and identification of plant genes able to confer salt tolerance upon overexpression. 

## 2. Functional Complementation as an Approach to Identify and Characterize Plant K^+^/Na^+^ Channels and Transporters

The functional complementation approach has been extremely successful for the identification and molecular cloning of plant ion channels. In 1992, the first two inward rectifying plant K^+^ channels (KAT1 and AKT1) were isolated by functional complementation of a yeast mutant devoid of its high affinity K^+^ transporter genes [15,16]. This seminal work set the paradigm for this experimental approach. Since then, several K^+^ transporters and regulators have been characterized, not only from plants, but also from mammals, viruses and bacteria [20,21,26,27,28,29,30,31,32,33].

A brief summary of the major contributors to K^+^ uptake and Na^+^ extrusion in yeast will be useful for understanding the details of the genetic backgrounds that are exploited in the identification and subsequent functional studies of heterologous ion channels and transporters (Figure 1). For an extended description of the mechanisms and regulation of Na^+^ and K^+^ transport and homeostasis in yeast, we refer the reader to a comprehensive review [34].

Nutritional uptake of K^+^ in *S. cerevisiae* depends mainly on two K^+^ transporters, named Trk1 and Trk2 [35,36,37]. These transporters use the electrochemical gradient generated by the plasma membrane H^+^-ATPase encoded by the *PMA1* gene to mediate high affinity uptake against the concentration gradient accumulating concentrations of approximately 200 mM in the cytosol even when the external concentration is as low as 10 µM. Trk1 contains 1235 amino acids and has been proposed to contain four repetitions of an M1PM2 motif based on its homology to the KcsA K^+^ channel from *Streptomyces lividans* [38]. M1 and M2 are transmembrane segments that are connected by the P helix (Figure 2). Residues in the second transmembrane helix (M2) of the fourth M1PM2 repetition (M2D) have been shown to be crucial for Trk1-mediated K^+^ transport [39]. Structural prediction models suggest that the Trk1 monomer assembles into a dimer or possibly a tetramer, which would lead to the formation of a “metapore” that could be responsible for Cl^−^ currents that have been observed in electrophysiology experiments [38,40,41]. Trk2 encodes a protein that is 55% identical to Trk1 [37], sharing the same topology, but differing in the length of the second cytosolic segment, which is considerably shorter in Trk2 (Figure 2). Trk1 and Trk2 allow yeast cells to grow under low K^+^ conditions and low pH. Trk1 is largely responsible for high affinity K^+^ influx, but is not considered as essential since the *trk1* simple mutant and even the *trk1 trk2* double mutant can grow in media supplemented with millimolar concentrations of K^+^. Deletion of the *TRK2* gene has little effect on yeast growth on its own but increases by 10-fold the concentration of K^+^ required for growth of the *trk1 trk2* double mutant [37]. Thus, the *trk1 trk2* mutant cannot grow in limiting K^+^ concentrations (below 1 mM). This phenotype can be rescued by the heterologous expression of different types of functional K^+^ channels, as will be discussed below. 

As previously described, Trk1 and Trk2 are responsible for high affinity K^+^ uptake. These transporters have been shown to be relatively specific for K^+^ vs. Na^+^ ions [42]. However, under conditions of salt stress, Na^+^ can enter the cell through these transporters and through other non-specific transporters (reviewed in [43]). Under conditions of high external K^+^ or Na^+^ concentrations, two major transporters are responsible for the extrusion of these ions across the plasma membrane. The *ENA1* gene encodes for a P-type ATPase whose expression is markedly induced upon salt or alkaline stress (reviewed in [44]). ENA transporters are localized to the plasma membrane and use the energy generated from ATP hydrolysis to extrude K^+^ or Na^+^ out of the cell [45,46,47]. Most yeast genomes contain three to five tandem copies of the ENA ATPases, thus requiring the deletion of the entire cluster to eliminate this cation extrusion function (for reviews, see [43,48,49]). At acidic pH, a second Na^+^/K^+^ extrusion system becomes important for yeast salt tolerance, the Nha1 antiporter. Nha1 is localized at the plasma membrane and acts as a dimeric, electrogenic proton antiporter with similar affinity for both K^+^ and Na^+^ [50,51,52]. Therefore, strains lacking the ENA cluster and the *NHA1* gene are highly salt sensitive and have been used extensively to identify and characterize plant genes involved in cation extrusion and salt tolerance in general, as will be discussed in the upcoming sections. 

Tok1 is the only known outward rectifying K^+^ channel in yeast [53]. It is a plasma membrane protein and its activity contributes significantly to the maintenance of the membrane potential [54]. However, the *tok1* simple mutant does not have obvious growth phenotypes in high or low K^+^ media, but the mutant is tolerant to killer toxins [55].

Additional K^+^ transporters are present in the yeast vacuolar membrane. This category of transporters includes Vnx1 and Vhc1. Vnx1 works as an antiporter, exchanging vacuolar protons for cytosolic K^+^ or Na^+^. Accordingly, Vnx1 is involved in Na^+^ compartmentalization and thus, cytosol detoxification [56]. The *VHC1* gene encodes for a vacuolar transporter that participates in K^+^ homeostasis [57]. It functions as a vacuolar K^+^/Cl^−^ symporter, contributing to the maintenance of intracellular K^+^ concentrations and vacuole morphology [57,58].

In addition, there are several ion transport proteins in other compartments and organelles. For example, in the endosome membrane of yeast cells, there is a K^+^/Na^+^ transporter, Nhx1. It is a H^+^ antiporter that operates specifically compartmentalizing the cations and extruding H^+^ to the cytosol. Together with Vnx1, the Nhx1 transporter contributes to the vacuolar and endosomal sequestration of excess cations present in the cytosol, thus maintaining luminal pH of the vacuole and endosomes, respectively [59]. Kha1, works as a K^+^/H^+^ antiporter in the Golgi apparatus [60] and Mkh1 constitutes the system of exchanging protons for K^+^ in the mitochondria. Mkh1 is essential for mitochondrial homeostasis and consequently for respiratory growth of yeast cells [61]. 

## 3. Milestones in the Identification of K^+^ Channels and Transporters in Plants 

The *trk1 trk2* mutant has been extensively used in several studies involving K^+^ channel identification, characterization and regulation, not only for plant genes, but also for mammalian and bacterial ion transporters. As mentioned, Trk1 transporters use the electrochemical gradient generated by the Pma1 H^+^-ATPase to mediate high affinity K^+^ uptake. In fact, these transporters are the major consumers of this electrochemical gradient [62]. The membrane potential of the *trk1 trk2* double mutant is more negative than the wild type strain, presumably because the positive charges in the form of protons extruded by Pma1 are not efficiently replaced by K^+^ ions via the Trk transporters [63]. Thus, this increased negative potential allows for intracellular K^+^ accumulation through functional heterologous K^+^ channels, which can be easily monitored by growth assays in low K^+^ media. 

The identification of new genes involved in K^+^ uptake (K^+^ channels and transporters) has been mainly based on yeast mutant complementation assays. Briefly, in this procedure a cDNA library (or a candidate gene) from a specific organism is cloned into a yeast plasmid and subsequently transformed into a yeast strain that can be used to select for the function of the transport protein. Table 1 lists the yeast strains lacking the *TRK1* and *TRK2* genes that have been used in this type of approach. The identification of positive clones is rapid and with minimal bias. The expression of a functional, heterologous K^+^ channel rescues the poor growth phenotype of the mutant strain under limiting K^+^ concentrations. In the specific case of plasma membrane localized transporters, the genetic background preferentially used for the screening is the *trk1 trk2* double mutant. This strain has also been employed in other studies, whenever the slow growth phenotype readout in low K^+^ media is useful. 

Here, we will describe in more detail a few major examples of plant K^+^ channels and transporters identified by yeast complementation. As previously mentioned, the first plant K^+^ channel cloned was KAT1, an inward rectifying channel primarily involved in stomatal opening. KAT1 was cloned by functional complementation of the high affinity K^+^ transport-defective *trk1 trk2* mutant by screening an Arabidopsis cDNA library. The expression of AtKAT1 in the yeast rescued growth on 50 µM KCl, where the mutant normally requires at least 50 mM KCl for growth [64]. Subsequently, the injection of rRNA encoding AtKAT1 into *Xenopus* oocytes confirmed an inward K^+^ current, displaying a pattern characteristic of an inward rectifying channel with a high selectivity for K^+^ ions [27]. KAT1 belongs to the Shaker-family of transporters and resides in the plasma membrane of guard cells. This category of K^+^-selective channels typically consists of six trans-membrane domains, a cyclic nucleotide binding domain, a KHA domain rich in hydrophobic and acidic residues and some have an ankyrin repeat domain (Figure 2) [4,15,27,65,66,67,68]. Based on the structural data obtained from bacterial and animal Shaker channels, this family has been proposed to adopt a tetrameric pore forming structure, which can be composed of homo or hetero-tetramers [69,70]. In particular, KAT1 is able to form hetero-tetrameric channels at the guard cell plasma membrane with KAT2 [71]. Further characterization of the key KAT1 structural components has been carried out by random mutagenesis and complementation of the yeast *trk1 trk2* mutant. These studies helped to define the positions in the pore that are determinant for K^+^ selectivity [72,73,74,75,76]. KAT1 isoforms from other plant species, such as potato [21], maize [77] and rice [78] have also been identified and functionally characterized.

The AKT1 channel was isolated almost simultaneously to KAT1 [16]. The strategy adopted was the same: Functional complementation screening of an Arabidopsis cDNA library in the *trk1 trk2* yeast strain. Neither KAT1 nor AKT1 are homologous to the yeast *TRK1* gene product, but both are able to rescue the slow growth phenotype of the mutant under limiting K^+^ conditions. AKT1 also belongs to the Shaker family and shares sequence homology with KAT1, but it is mainly expressed in roots and has been shown to be largely responsible for K^+^ uptake from the soil. The complex kinetics of K^+^ uptake observed in the complemented yeast mutant were in good agreement with those observed for K^+^ root transporters, corroborating this conclusion [16,79,80,81,82]. The selective pore positions in this channel were later characterized by random mutagenesis and functional complementation in yeast, again using the *trk1 trk2* mutant [83]. AKT1 homologues were subsequently isolated from higher plant species, such as barley (HvAKT1), maize (ZMK1), wheat (TaAKT1), potato (SKT1) and tomato (LKT1), and it was demonstrated that the function, and to varying extents, the sequence, are conserved, as well as the plasma membrane localization in root cells. In these cases, the characterization experiments were carried out in other heterologous non-yeast systems [28,84,85,86,87,88]. 

HAK genes encode for high-affinity K^+^ transporters expressed in root cells, where they contribute to the K^+^ uptake from the soil. The first HAKs isolated in plants were obtained from barley root cDNA, using an RT-PCR approach based on amino acid homology between the sequences of two K^+^ transporters known to belong to the HAK family, Kup of *Escherichia coli* [89] and HAK1 of *Schwanniomyces occidentalis* [90]. The characterization of these genes (HvHAK1 and HvHAK2) was carried out in *trk1 trk2* yeast mutants, where the authors concluded that only HvHAK1 was able to rescue the growth phenotype, confirming the high affinity K^+^ transporter function [91]. 

AtHAK5 shares structural and functional characteristics with AKT1 [13,92]. This gene encodes a transporter cloned from Arabidopsis using degenerate primers deduced from the conserved regions DNG(D/E)GGTFA and FADLGHF, respectively present in the HAK1 sequence previously cloned from *Hordeum vulgaris*, *HvHAK1* [91]. After cloning by RT-PCR, the full-length cDNA was inserted into a yeast expression vector and its functionality was tested by complementation of the growth defect of the *trk1 trk2* mutant [93]. Other AtHAK family members have been cloned and tested for yeast complementation, but they do not rescue the growth phenotype of the yeast K^+^-deficient mutant. 

Another category of K^+^ transporters identified using the yeast model system is represented by the *KT/KUP* family. They include *AtKUP1* [26,94], *AtKT2/KUP2* [95] and *AtKT3/KUP4* [96], which were isolated by screening an Arabidopsis cDNA library in the *trk1 trk2* mutant strain and selecting for growth under low K^+^ conditions. Analysis of the tissue expression of these transporters revealed that KT/KUP transporters are expressed in roots and in aerial parts of the plant [80].

KUP1 seems to work in both high- and low-affinity modes, which is characteristic of plant root K^+^ transporters, such as AKT1. The transition from the high-affinity to the low-affinity mode occurs at 100 µM to 200 µM external KCl. Given that K^+^ uptake via *AtKUP1* was inhibited by NaCl and other K^+^ channel blockers, the authors suggested that KUP1 was possibly participating in K^+^ uptake from the soil. However, AtKUP1 could not be expressed in *Xenopus* oocytes, leaving the molecular characterization of the currents mediated by this channel unresolved [94]. In addition, we were not able to find any other study on this transporter in literature, thus the mechanism of K^+^ uptake by AtKUP1 seems to be still undefined. 

AtKT2 mediates low-affinity K^+^ transport, likely facilitating passive diffusion of K^+^, while AtKUP4 mediates high-affinity K^+^ uptake. Recently, it has been reported that KUP4 is also involved in the process of embryo development during seed maturation [97].

High-affinity K^+^ transporters (HKT) transporters are structurally related to fungal and bacterial K^+^ transporters from the Trk/Ktr families. Phylogenetic and functional analyses indicate that HKT transporters can be classified into two sub-groups: Group I HKT transporters that are mainly Na^+^ selective, while group II HKT transporters can operate as Na^+^-K^+^ symporters or as K^+^-selective uniporters. Selective permeability to K^+^ relies on the glycine residues present in the middle of the pore of HKT/Trk/Ktr transporters. In fact, a substitution of at least one of the conserved glycine residues for a serine abolishes the selectivity to K^+^ and renders these transporters Na^+^-selective [98,99]. The HKT proteins are of particular interest to plant biologists, as some members of this class play an important role in salinity tolerance [100]. The first HKT1 gene characterized as a Na^+^-K^+^ symporter using the yeast functional complementation strategy in the *trk1 trk2* mutant was from wheat [101]. Detailed molecular studies revealed that the transporter presents specific binding sites for each of the ions. In particular, when K^+^ and Na^+^ are present at similar concentrations, HKT1 functions as a Na^+^-coupled K^+^ transporter. Nevertheless, when Na^+^ is present at toxic concentrations (millimolar), the ion binds to the high-affinity K^+^-coupling sites, resulting in a low-affinity Na^+^-Na^+^ uptake [102,103,104]. The fact that Na^+^ competes with K^+^ when it is present at high concentrations suggests that HKT1 may be one of the Na^+^ transporters in plant roots, which is relevant for salt toxicity in plants. A genetic selection of mutations of HKT1 revealed some relevant positions conferring salt resistance to yeast and oocytes; for instance, N365S and Q270L are the most effective. The approach used in this study was based on yeast complementation of *trk1 trk2* and *ena1-4 trk1 trk2* strains [105].

In plants, it was shown that HKT1 is present in the plasma membrane of xylem parenchyma cells and participates in the extraction of Na^+^ ions from xylem vessels, thus contributing to the salt tolerance response [106]. The majority of the HKT transporters characterized later from other plant species (i.e., eucalyptus, rice, wheat) are all permeable to Na^+^ as well [107,108,109]. It is interesting to note that not all the HKTs isolated are able to rescue the yeast K^+^-deficiency phenotype. For instance, two out of the five HKT cDNAs isolated from rice (OsHKT1 and OsHKT4) were confirmed to be Na^+^-specific by functional complementation of the *trk1 trk2* mutant [110]. 

More recently, an Arabidopsis AKT1-like channel was cloned from *Lilium longiflorum* pollen (*LilKT1*). The authors described the identification of this sequence from a pollen cDNA library by PCR amplification using degenerate primers derived from conserved regions of K^+^-channels. Ion currents are extremely relevant for pollen tube growth and thus for ovule fertilization. *LilKT1* was subsequently characterized using the yeast functional complementation assay [111]. *LilKT1* was expressed in two different yeast mutant strains lacking the *TRK1* and *TRK2* genes. Although overexpression of this channel did not fully complement the growth of these mutant strains, the authors demonstrated that this was because the K^+^ channel was unable to efficiently reach the plasma membrane in this heterologous system. Until the appearance of this report, the only inward rectifying K^+^ channel characterized in pollen grain was AtAKT6, which was characterized using the classical patch-clamp approach [22,112].

For studies of K^+^ channels that are not localized at the plasma membrane, alternative yeast mutant strains can also be employed. For example, yeast strains lacking the *YVC1* gene have been used to study the function of TPKs (also known as KCO), which are tonoplast K^+^ channels [113,114]. The TPKs are a two-pore K^+^-channel family, with four transmembrane and two pore domains. They localize mainly in the vacuolar membrane in yeast and in the tonoplast in plants [115,116], with the exception of *AtTPK4*, which has been localized in the pollen tube plasma membrane. This channel contributes to K^+^ conductance, as demonstrated by the complementation of the K^+^-transport deficient yeast mutant PLY246, carrying *trk1*, *trk2* and *tok1* mutations [117]. In the case of the vacuolar channels, to use the yeast functional complementation approach, plant genes are expressed in *yvc1* mutants and their activity is analyzed in isolated vacuoles. Using this strategy, the *AtTPK1* gene product showed ion channel activity expected for an SV-type channel, with a strong selectivity for K^+^ over Na^+^ [113]. In addition, a TPK2-like transporter from *Nicotiana tabaccum* (*NtTPK1*), isolated by querying a tobacco database for sequences homologous to the *AtTPK*-family, was functionally characterized by whole cell patch clamp recordings on isolated vacuolar membranes from the yeast strain SH1006 lacking the *YVC1* gene [114].

By using yeast strains defective in *nhx1*, the *A. thaliana NHX1* homologue, *AtNHX1,* was characterized. The sequence was deduced from homologies with *ScNHX1* and cloned using degenerate primers [118]. *AtNHX1* complemented the yeast *nhx1* mutant as shown by suppression of its extreme sensitivity to hygromycin and NaCl under conditions in which the K^+^ availability was reduced [119]. Subsequently, the Na^+^/H^+^ antiporters AtNHX1/2/5 and AtNHX6 were cloned and have been described as regulators of growth, flower development and reproduction, through the control of vacuolar pH and K^+^ homeostasis [120,121].

A few other examples of NHX genes identified in higher plants using yeast mutants have been described in the literature. For example, in 2002, based on the sequence homologies of Nhx1 cloned in yeast, rat, human, nematode and Arabidopsis, *BvNHX1* (from *Beta vulgaris*) was cloned [122]. It was able to complement the *nhx1* mutation in yeast, recovering the salt sensitive phenotype of the strain. In planta, mRNA transcript and protein levels correlated with an increase in vacuolar Na^+^/H^+^ antiporter activity in response to salinity treatment. With the same approach, *LeNHX2* was cloned from tomato [18]. The gene was able to rescue the salt and hygromycin sensitivity phenotype of the yeast *nhx1* mutant. It was confirmed that *LeNHX2* mediates K^+^/H^+^ exchange, and to a lesser extent, Na^+^/H^+^ exchange in vitro. In addition, the NHX1 protein from *Eutrema salsugineum,* formerly *Thellungiella halophilae*, has also been functionally expressed in yeast [123]. 

The utility of yeast in the study of K^+^ channels not only includes the discovery of new heterologous K^+^ channels. In fact, there are examples in which the recovery of the phenotype of a yeast mutant defective in high affinity K^+^ transport has been used as a “proof of concept” for newly engineered K^+^ channels. This is the case of BLINK1, a blue light-gated K^+^ channel engineered by connecting the light sensor module of *Avena sativa* phototropin LOV2 to the N-terminus of Kcv, the smallest K^+^ channel known from chlorella virus [124,125]. The authors created a new method to combine a yeast-based screening system with light-activated K^+^ conductance. Finally, yeast has also been used as a chassis to improve the salt vacuolar accumulation capabilities of NHX1 by directed evolution using random mutagenesis and DNA shuffling [126].

## 4. High-Throughput and Directed Protein-Protein Interaction Assays Used to Identify Plant K^+^/Na^+^ Transporter Regulators

As reviewed in the previous section, yeast represents a powerful tool to study plant ion channels and transporters. Yeast can also be used as a high-throughput platform for the identification and subsequent study of protein–protein interactions (PPIs). Here we describe the state-of-the-art of the available PPI techniques and discuss the K^+^Na^+^ transporters/channels interactors identified using these approaches.

## 5. State-of-the-Art of the Available Techniques for Detecting Protein–Protein Interactions in Yeast

Since the classic yeast two-hybrid (Y2H) assay, new techniques to detect PPIs have been developed. These new techniques solve some of the limitations of the previous ones and offer a broader spectrum of possible baits or proteins that can be analyzed, mainly, according to their topology or subcellular localization. A summary of the important features of these techniques is shown in Table 2.

Basically, all these techniques are different variants of functional reconstitution assays, where a reporter protein is separated into domains or fragments and upon interaction of the test proteins, the function of the reporter protein is reconstituted. The Y2H system has, without doubt, represented a revolution in the field, allowing for the detection of interactions in vivo in a true cellular environment. The classic Y2H technique is currently the most commonly used method for investigating PPIs. 

Among the different variants, the ones that are preferentially used in the plant sciences are the classic Y2H and the membrane split-ubiquitin system (reviewed in [151]). Below, we describe briefly selected Y2H variants. More detailed descriptions are provided in several publications [152,153,154]. A schematic diagram of these techniques is shown in Figure 3.

### 5.1. Classic Y2H System (Y2H)

In this system, one protein is fused to the DNA-binding domain of a transcription factor, while the other protein is fused to a transcriptional activation domain [127]. Interaction between the bait and prey proteins reconstitutes the function of the transcription factor, leading to transcriptional activation of a reporter gene. Interacting proteins are identified by growing the yeast cells under conditions where cell survival is dependent on the transcription of the reporter gene. The most common reporter genes are *HIS3* and *ADE2*, which reconstitute histidine and adenine biosynthesis, respectively. The bacterial *lacZ* gene is also commonly used as a reporter gene. In this case, upon interaction expression of the *lacZ* gene leads to the accumulation of the β-galactosidase enzyme that produces a blue product when X-gal (5-bromo-4-chloro-3-indolyl-β-d-galactopyranoside) is added to the media.

It is also possible to first transform a set of different bait constructs in one mating type and a set of preys in the other mating type; then, by generating diploids by mating the two haploid strains, large numbers of binary interactions can be evaluated by growing in selective media (reviewed in [152]).

Limitations: Proteins have to enter the nucleus in order to induce the expression of a reporter gene; many false positives and negatives; lack of interaction dynamics and transactivating proteins cannot be properly analyzed. The reasons behind these shortcomings and some minor modifications to solve them have been recently reviewed and some of them will be discussed below [152]. 

### 5.2. Reverse Y2H System (rY2H)

This system is based on the classic Y2H assay, but in this case the interaction of two proteins leads to expression of a gene that confers toxicity to the yeast cells. The most commonly used is the *URA3* gene that encodes a decarboxylase which produces a toxic intermediate in media containing 5-Fluoroorotic acid (5-FOA) leading to yeast cell death. This assay is used to identify inhibitors of protein–protein interactions or to select for mutants encoding proteins that have lost their ability to interact, thus providing a high-throughput system to identify inhibitors and key residues mediating protein–protein interactions. Removal of 5-FOA converts this assay to classic Y2H (reviewed in [152]).

Two additional systems using *CYH2* or *GAL1* instead of *URA3* plus 5-FOA have been developed (reviewed in [152] and [153]). In the first case, a mutant background containing a cyclohexamide-resistant *cyh2* allele is used. The interacting proteins drive the expression of the wild type copy of *CYH2*, which promotes sensitivity to cycloheximide. In the second system, the screening takes place in a strain that lacks the *GAL7* gene, which catalyzes the conversion of galactose-1-phosphate into glucose-1-phosphate. The protein–protein interaction drives the expression of the galactokinase-encoding *GAL1* reporter gene. In this system, when the protein–protein interaction occurs, galactose accumulates and reaches toxic concentrations [155]. 

Application: Detection of inhibitors or essential amino acids for the PPI. 

### 5.3. Repressed Transactivator System (RTA) 

The principle is similar to the rY2H system, however, instead of a transcriptional activation domain, the prey protein is fused to a transcriptional repression domain. Interaction of the two proteins leads to repression of the *URA3* gene and consequently resistance to media containing 5-FOA [132]. This system was extended to screen for inhibitors of protein–protein interactions or to select for mutants encoding proteins that have lost their ability to interact. In this case, interaction of the two proteins leads to repression of a positive selection marker (reviewed in [154]). For example, upon interaction of bait and prey, the transcriptional repression domain of Tup1 would lead to repression of *HIS3* and consequent growth deficiency on medium without histidine.

Advantages: The original version eliminates false positives by proteins that are transcriptional activators. The alternative versions can be used to identify inhibitors or essential amino acids for the PPI.

### 5.4. RNA Polymerase III System (Pol III)

This technique also derives from the classic Y2H system, however, the second protein is fused to Tfc3 (aka τ138), a subunit of the multimeric protein complex TFIIIC (one of the two transcription factors involved in RNA polymerase III-mediated transcription). After interaction of both proteins, the TFIIIC complex is bound to DNA and recruits a second transcription factor (TFIIIB) and Pol III. This will activate transcription of the reporter gene (reviewed in [153,154]).

Advantages: Elimination of false positives by proteins that are able to autoactivate RNA pol II on their own.

### 5.5. Small-G-Protein-Based Methods

These methods make use of the Ras pathway, which is essential in yeast. They are based on the reconstitution of the function of the Ras pathway by bringing the Ras-GEF protein, SOS or Ras itself to the membrane for activation. 

-SOS recruitment system (SRS): One protein is fused to a modified human Sos protein (hSos), which can functionally replace its yeast counterpart Cdc25, but only if it is targeted to the plasma membrane. The other protein is fused to the C-terminus of the v-Src myristoylation sequence, which targets proteins to the membrane. Interactions between both proteins leads to the recruitment of hSos to the membrane and activation of the yeast Ras pathway that complements the temperature‑sensitive *cdc25* mutation at the restrictive temperature (36 °C) [139]. Advantages: Can be applied to cytosolic proteins that are unable to enter the nucleus or that require post-translational modifications in the cytoplasm.-Ras recruitment system (RRS): The principle is similar to the SRS, however, hSos is substituted by a mutant form of mammalian Ras (mRas). This Ras protein (Ras(61)ΔF), is a constitutively active form of mammalian Ras, which lacks the CAAX box required for its lipid modification and subsequent localization to the plasma membrane [140]. The bait protein is fused to this version of mRas. The prey proteins are fused to a membrane localization sequence. In this way, if the bait and prey proteins interact, the constitutively active form of Ras is recruited to the plasma membrane and can complement the *cdc25* temperature-sensitive mutant.Advantages: Reduction of false positives; furthermore, the smaller size of mRas compared with hSOS reduces the steric hindrance problem observed with the large hSos protein.-Reverse Ras recruitment system (rRRS): The principle is similar to the RRS, however, mRas is fused to the prey protein and the bait protein contains its own membrane localization signal or is an intrinsic membrane protein [138].Advantages: Can be applied to membrane proteins.

### 5.6. Heterotrimeric G-Protein Fusion System

Interaction between integral membrane bait and a soluble prey protein, fused to the γ-subunit (Ste18) of a heterotrimeric G-protein, will sequester the β-subunit, thus disrupting formation of heterotrimeric G-protein complex and subsequent downstream signaling, leading to growth arrest in a pheromone-dependent growth inhibition assay (halo assay) and reduced expression of a pheromone-controlled reporter in a *ste18* strain [137]. 

Advantages: Can be applied to membrane proteins; furthermore, only one of the two proteins is a fusion protein.

### 5.7. Screening for Interactions Between Extracellular Proteins (SCINEX-P) 

The two proteins to be tested are N-terminally fused to the endoplasmic reticulum (ER) transmembrane protein mutants Ire1_K702R_ and Ire1∆tail, respectively. Interaction between the two proteins, leads to dimerization of both Ire1 mutant forms at the ER membrane and functional complementation of the Ire1 dimer, which is the principle mediator of the unfolded protein response (UPR) in yeast. Upon Ire1 activation, the Hac1 transcription factor is activated and, in this system, leads to the transcriptional activation of reporter genes containing Hac1-responsive promoters [143].

Advantages: Can be applied to proteins located in the lumen of the ER, including extracellular and secreted proteins.

### 5.8. Golgi Y2H System (GY2H) 

The two proteins of interest are fused to both modular domains of the Golgi-resident glycosyltransferase Och1: The N-terminal LOC domain for membrane attachment and the C-terminal CAT domain that performs the mannose transfer reaction within the Golgi lumen. This is an essential reaction for the production of the high mannose structure that covers the cell wall of wild-type yeast. The absence of this mannose structure renders yeast cells sensitive to high temperatures and to cell wall damaging agents, such as the benzidine-type dye Congo red. Upon interaction of both proteins, the function of the Och1 protein is reconstituted and the *och1* mutant will be able to grow at the non-permissive temperature (37 °C) or in the presence of Congo red [144].

Advantages: Can be applied to proteins located in the Golgi lumen. It has been used to characterize interactions between transactivating transcription factors that cannot be studied using traditional Y2H approaches.

### 5.9. Dual-Bait System

The principle is similar to the classic Y2H system, however, two known interactors of a protein of interest are each fused to a different DNA-binding domain, each of which targets the promoter of a different reporter, while the (mutated) prey protein is attached to an activation domain [131]. With this system, mutations that specifically target only one interaction, leaving the other interaction intact, can be identified.

Application: Analysis of specific binding sites for proteins with multiple interactors.

### 5.10. Split-Ubiquitin System 

In this system, the protein that is reconstituted is ubiquitin. The carboxy-terminal fragment of ubiquitin (Cub) is fused in frame to a fully functional yeast transcription factor, which is in turn fused to a protein that is excluded from the nucleus (usually an integral membrane protein). The prey protein is fused to a mutated version (I13G) of the N-terminal region of ubiquitin (NubG), which has very low propensity to bind to the Cub fragment. Upon protein–protein interaction, the ubiquitin protein is reconstituted, and ubiquitin-specific proteases cleave the transcription factor from the Cub fusion, which then travels to the nucleus to activate the transcription of reporter genes.

-Membrane split-ubiquitin system (MbY2H): The bait protein needs to be excluded from the nucleus and the topology must be such that the fusions are in the cytosol. This system has been successfully applied to integral membrane proteins. It can also be used for proteins that are resident in other membrane systems or those that have lipid modifications [135]. Advantages: Can be applied to membrane proteins.

Moreover, mating type a and alpha two-hybrid strains have been developed that enable very efficient mating-based screenings using yeast strains already containing high coverage cDNA libraries from different organisms [156]. The prey protein is cloned into the appropriate Cub vector and transformed into the correct mating type, which can then be mated with high efficiency to a strain of the opposite mating type containing Nub-fused cDNA libraries.

-Cytosolic split-ubiquitin system (CytoY2H): The same strategy as that used for the MbY2H system is employed but, in this case, the bait protein does not have to be a membrane-bound protein by itself, because the integral ER membrane protein Ost4 is added at the N-terminal end of the bait to impede its entry into the nucleus (reviewed in [152,153,154]).Advantages: Can be applied to cytosolic proteins that require post-translational modifications in the cytoplasm.-Generally applicable split-ubiquitin system: In this case, the transcription factor is replaced by the reporter protein Ura3, with an arginine residue (R-Ura3) between Ura3 and Cub. After interaction of bait and prey, which can reside in membranes or the cytosol, Ura3 is cleaved off and it is quickly degraded due to the exposed N-terminal arginine residue. Consequently, the cells become resistant to 5-FOA. The R-Ura3 method is especially suitable for finding transcription factor partners, both activators and repressors (reviewed in [153]).Advantages: Careful optimization of 5-FOA levels reduces false discovery rates.

Application: Studies on protein conformations changes. Significant changes in the conformation of a protein can be expected to alter the distance between the N-terminus and the C-terminus of most proteins. By attaching Nub to the N terminus and Cub to the C terminus of a single polypeptide, changes in the time-averaged distance between the N and the C terminus can be measured according to the efficiency of the Nub-Cub reconstitution [157].

### 5.11. Split-Trp System

The two proteins are fused to the C-terminal (CTrp) and the N-terminal (NTrp) fragments of the Trp1 protein required for tryptophan biosynthesis. Upon interaction, a quasi-native Trp1 protein is reconstituted and allows *trp1* deficient yeast strains to grow on medium lacking tryptophan (reviewed in [154]). 

Advantages: Can be applied to all types of proteins, independently of their subcellular localization.

### 5.12. Split-mDHFR System

The two proteins are fused to two dihydrofolate reductase (DHFR) fragments. After interaction of both proteins, DHFR is reconstituted and catalyzes the reduction of dihydrofolate into tetrahydrofolate, which is essential for cell proliferation, growth and survival. 

The murine DHFR (mDHFR) can serve as a reporter protein in bacterial and fungal DHFR systems in which a PPI is detected by survival of the cell in the presence of methotrexate or trimethoprim, with mDHFR taking over the function of the host DHFR protein [146]. 

Advantages: Can be applied to all types of proteins, independently of their subcellular localization.

As reviewed in [152], new challenges are to devise novel yeast systems which can provide real-time data and more reliable outputs. It may be useful to design alternatives that can report PPIs more specifically and sensitively than yeast growth phenotypes, but with the same level of simplicity. In this sense, fluorescence-based systems for high-throughput screening in yeast could be an alternative, although adaptation for single cell detection and automation will be required for these types of techniques to compete with current growth-based selection systems [158].

## 6. Interactors of K^+^/Na^+^ Transporters/Channels Detected Using Protein–Protein Interaction Techniques in Yeast

Many PPIs have been described for K^+^/Na^+^ transporters/channels in plants (reviewed in [159]). In this review, we will focus on those detected using the yeast-based methods described above (Table 3). 

Briefly, interactors have been detected for the three families of genes encoding plant plasma membrane K^+^ transporters: The HKT family, the HAK/KUP/KT K^+^ transporter family and the Shaker K^+^ channel family. Within the Shaker K^+^ channel family, interactors have been detected for almost all of the members. Specifically, interactors have been detected for HKT1, HAK1, KUP6, AKT1/2/3, KAT1/2 AtKC1, GORK, SKOR, as well as carrot KDC1, potato SKT2/3, OsKAT3 and subunits OsKOB1, KPutB1KST1 and TRH1.

Among the detected interactors of these K^+^ transporters/channels, several homo- or hetero-oligomerization events have been detected and are summarized in Table 4. 

Moreover, putative regulatory proteins such as kinases, phosphatases, GTPases, as well as anion channels, purine and nitrate transporters, targeting-related proteins and calcium sensors have been identified. The detected interactors and their proposed regulatory function are detailed in Table 5.

Indeed, at least two classes of kinases have been implicated in KAT1 channel regulation. For example, Open Stomata 1 (OST1, also called SnRK2.6 or SRK2E) is a Snf1-related kinase involved in abscisic acid (ABA) signaling that has been shown to phosphorylate KAT1 leading to channel inhibition in oocytes and yeast model systems [186]. These observations are in agreement with a more recent study showing a physical interaction between Ost1 and KAT1 (but not KAT2) in planta and Ost1-dependent regulation of ABA-mediated K_in_ currents in Arabidopsis guard cells [187]. 

Finally, two types of SNARE (soluble N-ethylmaleimide-sensitive factor protein attachment protein receptor) proteins, the Q-SNARE SYP121 and the R-SNARE VAMP721, have been shown to physically interact with KC1 and KAT1 and to modulate channel activity [164,165,166,188,189]. In this case, these proteins have been postulated to directly regulate KAT1 gating in opposing manners (SYP121 activates/VAMP721 inhibits) and, in fact, mutational studies have indicated that the trafficking and gating functions of SYP121 can be physically separated [164,189]. Moreover, primary evidence for the functional coupling of the SNARE SYP121 with Ca^2+^ channels underscores a mechanism for integration of SNARE- and Ca^2+^-mediated control of K^+^ channel currents in guard cells [189]. Furthermore, calcineurin B-like calcium sensors (CBLs) and a target kinase (CBL-interacting protein kinase 23 or CIPK23) appear to phosphorylate and activate AKT1 [168], and CBL4 mediates ER-to-PM translocation of AKT2 and enhances AKT2 activity upon interaction with CIPK6 although in a phosphorylation-independent manner [175].

As shown in Table 3, only classic Y2H and membrane split-ubiquitin techniques have been used as yeast-based techniques for the detection of PPIs of K^+^/Na^+^ transporters/channels in plants. Both techniques are limited to proteins capable of entering the nucleus and membrane proteins, respectively. In the future, the use of yeast-based techniques for detection of PPI of other type of proteins, such as those listed in Table 2, may reveal new interactors and expand the knowledge of K^+^/Na^+^ transporters/channel structure and their regulation.

## 7. Reconstitution of Functional Plant Ion Transport Systems in Yeast: The SOS Pathway Paradigm

In previous sections, we have summarized the use of yeast to clone and characterize ion transport proteins and yeast-based protein-protein interaction assays that have been employed to identify regulators of these channels and transporters. Another very powerful approach is the reconstitution of ion transport systems in yeast. In these cases, the transporter alone may have limited function and the co-expression of regulatory proteins is required to observe more robust phenotypes.

One canonical example of the use of yeast as a system to reconstitute a functional ion transport system is the SOS signaling pathway. The Arabidopsis *sos* (*salt overly sensitive*) genes were isolated in a genetic screen for plant mutants hypersensitive to NaCl [190]. SOS1 was the first plasma membrane Na^+^/H^+^ antiporter described in plants [191]. SOS2 is a Ser/Thr protein kinase, which belongs to the SnRK3 family and contains two functional domains [192]. SOS2 can also be found in the literature as calcineurin B-like protein-interacting protein kinase (CIPKs). SOS3, on the other hand, is a calcium sensor [193]. There is also a SOS4, involved in vitamin B6 metabolism [194] important for root hair development [195] and for vitamin-B6 mediated processes in the chloroplast [196].

Heterologous expression of SOS1-3 in yeast was instrumental in the characterization of the mechanism of action and the interplay among these proteins. SOS3 is a calcium sensor and was the first member characterized of the SCaBPs (SOS3-like calcium-binding protein) family, also named as calcineurin B-like/CBL. SOS3 recruits the SOS2 protein kinase to the plasma membrane upon activation by calcium. SOS2, once activated by SOS3, is able to phosphorylate SOS1 and increase its Na^+^ transport activity [197]. 

*S. cerevisiae* has demonstrated to be a very useful tool to reveal the molecular mechanisms governing the SOS system and their targets and specificities. As explained above, the complete system SOS1-3 can be reconstituted in yeast. SOS1 confers a weak phenotype of Na^+^ tolerance when expressed in yeast mutants lacking the endogenous Na^+^ extrusion systems (Ena1-4 and Nha1, see above). Overexpression of SOS2 and SOS3 in the yeast *ena1-4 nha1* mutant strain overexpressing SOS1 dramatically rescues the Na^+^ sensitivity phenotype, while overexpression of SOS2 or SOS3 in yeast in the absence of SOS1 does not confer any phenotype [197]. This explains why the SOS proteins have never been identified in a heterologous expression screening of plant genes in yeast. SOS1 overexpression confers a very weak phenotype and salt tolerance conferred by SOS2 and SOS3 requires SOS1. Nevertheless, yeast is an ideal platform that provides a fast and easy way to test putative SOS proteins. In the Arabidopsis genome, there are nine genes that encode putative SOS3 calcium sensors and 24 that encode putative SOS2 protein kinases [198]. This reconstitution system has facilitated the characterization of some of them as *bona fide* participants in the Na^+^ homeostasis machinery and has provided a system to determine their varying degrees of specificity and the combinatorial diversity of the complex. For instance, SOS2 may be activated by SOS3 or by its homologue SCaBP8/CBL10 [199], after the Ca^2+^-dependent binding to the FISL motif in the SOS2 NAF domain, which probably acts as an autoinhibitory domain. SOS3 is also phosphorylated by SOS2 [200] and this stabilizes the interaction and increases salt tolerance [201]. The SOS2/SOS3 module also activates the antiporter activity of the tonoplast NHX K^+^-Na^+^/H^+^ antiporters [202] and the H^+^/Ca^2+^ antiporter CAX1 [203]. The yeast model system also allows the cartography of the functional domains by using deletions or to analyze the effect of post-translational modifications like phosphorylation [204]. Therefore, the yeast reconstitution strategy has been a useful tool to identify and characterize plant proteins, which are upstream regulators of the SOS system, and therefore participate in the molecular response to salt stress [205].

Arabidopsis is the most popular model system for plant molecular biology, but yeast has also been used to characterize SOS proteins from other plants. In Table 6, the SOS orthologues whose function has been shown in yeast are summarized. 

It is also important to note that yeast is an optimal system to study the SOS pathway as the sodium extrusion ability of the SOS system, when heterologously expressed in yeast, is dependent on the proton gradient generated by the yeast proton pump Pma1 [218]. The SOS system would not likely be functional in an animal-based system, given that animal cells use sodium to energize the plasma membrane. In fact, yeast is also a good system to investigate the plant plasma membrane proton ATPase. In plants, the formation of the proton gradient depends on a multigene family [219]. In Arabidopsis, there are three major isoforms (AHA1–3). AHA2 was the first to be functionally expressed in yeast. The full version of the protein was retained in internal membranes and weakly complemented the growth defect of a conditional *pma1* mutant, but a truncated version in which the last 94 amino acids of AHA2 were deleted was able to partially localize to the plasma membrane and fully complemented the growth defect, suggesting an autoinhibitory function for this region [220]. The functional evaluation of the three isoforms in yeast indicates that the major differences among them are quantitative rather than qualitative, although the heterologous expression is not as efficient as other plant membrane proteins and most of it is retained in the endoplasmic reticulum [221]. The fact that AHA proteins are properly expressed and are functional in yeast enables the design of mutational approaches in order to identify versions with improved transport coupling efficiency [222]. 

Another example of reconstitution of a plant signaling pathway is related to ABA signaling. Upon abiotic stress, plants produce ABA. This hormone enters the cell and elicits plant responses by binding to soluble PYRABACTIN RESISTANCE1 (PYR1)/PYR1-LIKE (PYL)/REGULATORY COMPONENTS OF ABA RECEPTORS (RCAR) receptors, a family of proteins with 14 members. PYR/PYL/RCAR receptors perceive ABA intracellularly and, as a result, form ternary complexes and inactivate clade A PP2C phosphatases [223,224]. This activates downstream targets of the PP2Cs, like proteins belonging to the sucrose non-fermenting 1-related protein kinase (SnRK) subfamily 2, which regulate the transcriptional response to ABA and stomatal aperture. Additional targets of clade A PP2Cs include SOS2 members such as SnRK1, SnRK3s/calcineurin B-like (CBL)-interacting protein kinases (CIPKs), calcium-dependent protein kinases (CDPKs/CPKs), ion transporters such as the K^+^ channel AKT1 and AKT2 or the slow anion channel 1 (SLAC1) and SLAC1 homolog 3 (SLAH3), and transcriptional regulators, such as bZIP transcription factors and chromatin-remodeling complexes [225]. The point is that there is a multiplicity of possible interactions, as not all PYR/PYL/RCAR receptors interact with PP2C phosphatases and not all the interactions are ABA-dependent. In addition, C2-domain abscisic acid-related (CAR) proteins regulate the interaction of the hormone receptors with the plasma membrane [226]. Many of these interactions have been characterized by yeast two-hybrid, and the presence or the absence of the hormone in the yeast growth media can modulate this binding, therefore these ternary complexes (two proteins and ABA) can be reconstituted using yeast [227]. This system has been employed to carry out screenings to identify chemicals that can act as ABA agonists. These novel molecules may induce plant tolerance to abiotic stress by increasing ABA-mediated response [228]. 

## 8. Identification of Plant Genes Involved in K^+^/Na^+^ Homeostasis by Heterologous Expression 

As described above, a classical approach utilizing the yeast model system is to use mutant strains deficient in potassium uptake or in sodium extrusion and introducing plant cDNAs to complement a given phenotype, a strategy which is still in use [229]. This strategy is a way to identify transporters, to confirm their function and/or to perform molecular characterization by carrying out structure-function studies. This is possible because in yeast the quantification of the transport activity can be easily assayed by growth tests or defined in more detail by measuring ion transport and ion content. Another highly exploited approach is the use of yeast as a heterologous system to screen for plant genes involved in potassium and sodium homeostasis by screening for tolerance to salt stress induced by NaCl. The predecessor that set the paradigm for this strategy was a very successful screen carried out in the Serrano laboratory. In this case, instead of plant genes, yeast halotolerant (HAL) genes were identified by overexpressing a yeast genomic library and selecting clones able to grow in media with 1.2 M NaCl or 0.2 M or 0.4 M LiCl. In this screening procedure, the investigators identified Hal1 [230], Hal3, a regulator of the Ppz1 protein phosphatase, which regulates Trk1 and Trk2 [231], Hal4 and Hal5, two protein kinases that regulate Trk1 [232], Crz1 the calcineurin-regulated transcription factor, which controls the expression of *ENA1* [233] and Qdr2, an ABC multidrug transporter able to transport monovalent and divalent cations [234], among others. Some of these genes proved to confer salt tolerance when overexpressed in plants [235,236]. 

The key point regarding this technique is that it allows for the identification of genes related to K^+^/Na^+^ homeostasis. The main advantage of using this kind of approach is that it is fast and can identify genes previously unrelated to ion homeostasis. Since these initial studies using a yeast genomic library, this approach has been carried out by several groups using plant cDNA libraries (Table 7). There are several strategies to increase the chances of success, for instance, the cDNA library can be obtained from plants after a stress treatment to increase the number of transcripts related to the abiotic stress response. The screening can be performed in a wild type yeast or in a stress-sensitive mutant, such as those described above. The use of mutants is a way to enhance the sensitivity of the screening and increase the chances of identifying plant genes conferring tolerance. For genes conferring weak or mild phenotypes, the yeast stress defense system could mask the effect, but by using stress-sensitive mutants, functional genes can be isolated. The most typical strain for performing screenings is the *ena1-4* mutant that harbors a deletion in the genes encoding the Na^+^ extrusion pumps. 

One striking feature of this kind of screening is that most of the isolated genes are not conserved in yeast. One would expect that plant genes could confer salt stress tolerance in yeast by increasing K^+^ uptake, Na^+^ extrusion or Na^+^ accumulation in the vacuole, but what happens instead is that in many cases there is no change in the ion content. The tolerance is attained by complementing a molecular process, which becomes limiting upon a decrease in the K^+^/Na^+^ ratio, such as gene splicing, or by counteracting some of the side effects of the salt stress, like the osmotic stress induced by high concentrations of Na^+^ in the medium or the concomitant oxidation [237]. In Table 7, we have compiled the results found in the literature for plant genes isolated in heterologous screenings in yeast that confer salt tolerance upon overexpression, with information on the origin of the cDNA library, the yeast strain used and the screening conditions. This review is focused on the use of the baker’s yeast *S. cerevisiae*, but some laboratories have also used the fission yeast, *Schizosaccharomyces pombe,* in a similar manner with positive results [238].

The most represented category are genes related to protein post-translational modifications, including protein folding, sorting and degradation. This suggests that processes related to protein sorting or protein regulation are inhibited by salt stress or that plant proteins are regulating the yeast salt stress response proteins. For instance, among the genes identified in this type of screen is a cyclophilin [244] and an FKBP protein [256]. These proteins have peptidyl-prolyl cis-trans isomerase activity, and have been described to regulate the H^+^-ATPase in plants [261]. Another major category is RNA binding proteins. One of the few commercialized GMO crops with increased salt tolerance is the Droughtgard^®^ maize by BASF. The gene that confers this tolerance is a bacterial cold shock protein (CSPb from *Bacillus subtilis*), which also is an RNA binding protein. Therefore, this protein is likely to be a target of salt toxicity and overexpression of genes related to this process may complement the growth defect induced by salt stress [262]. Proteins related to protein translation, mainly ribosomal proteins, and photosynthesis-related proteins are also among the most common genes identified (Table 7). These are pivotal processes for plant cell physiology, so genes of both categories are abundant in cDNA libraries. It is likely that even cDNAs that confer a minor phenotype may be isolated due to the large number of individual cDNAs in the libraries. Finally, it is also interesting to point out that although some membrane transporters have been identified, Na^+^, K^+^ or Cl^−^ channels have never been recovered in these screenings. The chances of identifying a complex multimeric integral membrane protein are low, as not all plant membrane proteins are properly folded, processed, assembled or localized in yeast due to the absence of the specific machinery [263,264]. In addition, they may lose activity due to the different composition of the membrane lipids [265]. Among the identified membrane transporters there is only one protein annotated as a cation/calcium exchanger [260], PIP and TIP aquaporins [250,255] and an ABC multidrug transporter, among others [255]. In most cases, the phenotype is probably due to non-specific Na^+^ or K^+^ transport. 

It is also important to mention that heterologous expression of plant genes in yeast also has several limitations. For instance, not all plant genes are properly expressed in yeast due to codon-usage bias [266]. Another possible drawback of these kinds of screenings is the possibility of a dominant negative effect, so the phenotype observed may not be an indication of the function of the gene in plants, but represent an indirect effect due to interference with a yeast pathway that induces a phenotype similar to the one being evaluated [267]. These circumstances could explain the fact that in some screenings proteins related to signal transduction pathways, such as protein kinases [245] or ABA signaling-related proteins [260] also appear. Finally, it is also important to recall that at least some plant ion transport systems may also require multiple genes for complete function of the channel or transporter. The quintessential example here is the SOS system discussed above, which has never been identified by a cDNA library screening because only one gene is overexpressed at a time. Despite these shortcomings, the yeast heterologous expression strategy has made many important contributions to our understanding of plant K^+^ and Na^+^ homeostasis and will surely continue to provide valuable information in the future.

## Figures and Tables

**Figure 1 ijms-20-02133-f001:**
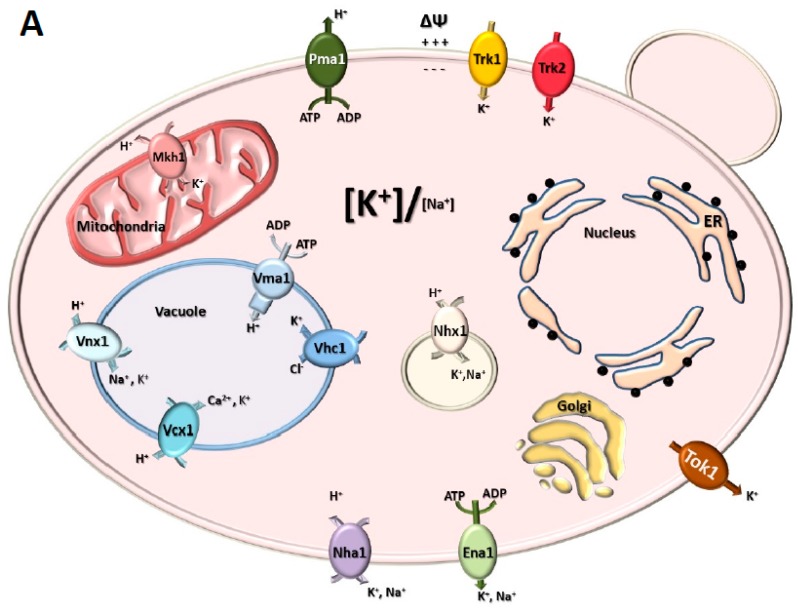
Schematic representation of the main monovalent channels and transporters in yeast and plant cells. (**A**) In a yeast cell, channels and transporters are present in almost all the organelles and cellular compartments. The introduction of positively charged ions and the expulsion of the negative ones maintains the negative plasma membrane potential. All the ion transporter proteins cited in the main text are represented. Inward/outward ion traffic is represented by arrows. (**B**) A schematic representation of a plant cell (without the cell wall). The KAT1 channel is represented in the known forms of homo-tetramer and hetero-tetramers with KAT2. All the transporters and channels cited in the text are represented. Organelle size is not to scale.

**Figure 2 ijms-20-02133-f002:**
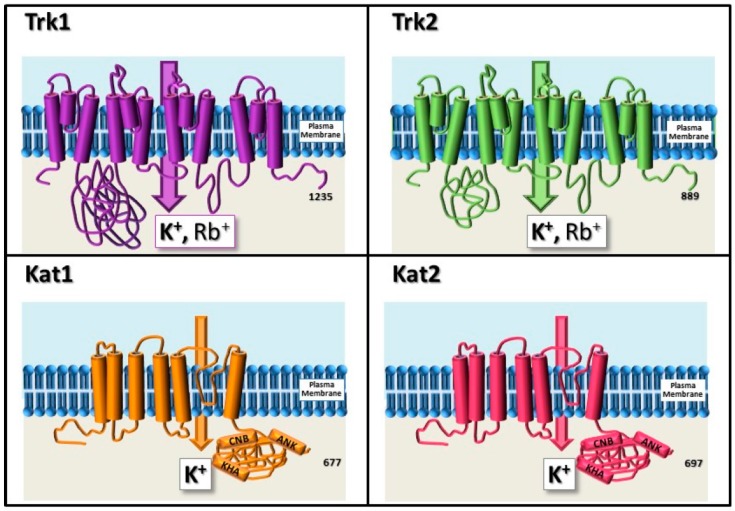
Schematic representation of selected K^+^ transporter proteins from yeast and plants. Top row: Yeast Trk1 and Trk2 transporters. The 4 M1PM2 structure is depicted. Bottom row: KAT1 and KAT2 channel monomers. CNB: cyclic nucleotide binding domain; KHA: KHA domain; ANK: Ankyrin repeat.

**Figure 3 ijms-20-02133-f003:**
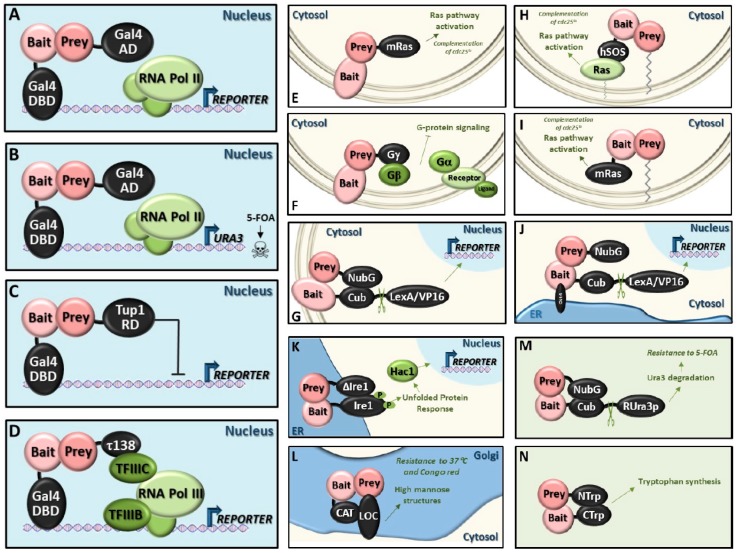
Diagram of the techniques for detecting protein–protein interactions in yeast. (**A**) Classic Y2H system. (**B**) Reverse Y2H system. (**C**) Repressed transactivator system. (**D**) RNA polymerase III system (the τ138 subunit is encoded by the *TFC3* gene). (**E**) Reverse Ras recruitment system. (**F**) Heterotrimeric G-protein fusion system. (**G**) Membrane split-ubiquitin system. (**H**) SOS recruitment system. (**I**) Ras recruitment system. (**J**) Cytosol split-ubiquitin system. (**K**) SCINEX-P system. (**L**) Golgi complex system (CAT: Catalytic domain Och1; LOC: Och1 domain required for membrane attachment). (**M**) Generally applicable split-ubiquitin system. (**N**) Split-Trp system. The subcellular location within a yeast cell and the operating mode (at the moment of bait-prey interaction) is represented. See text for more details.

**Table 1 ijms-20-02133-t001:** Yeast stains used for cloning and functional complementation of plant K^+^ channels. All strains lack the *TRK1* and *TRK2* gene.

Mutant Name	Genetic Background	Relevant Genotype
WΔ3	W303-1A	*MAT a ade2-1 canl-100 trpl-1 ura3-1 trk1::LEU2 trk2::HIS3*
CY162	R757	*MAT a ura3–52 his3Δ200 his44–15 trk1Δtrk2::pCK64*
9.3	W303-1A	*MAT a ena1-4Δ:HIS3::ena4Δ leu2 ura3–1 trp1–1 ade2–1 trk1Δ trk2::pCK64*
SGY1528	W303-1A	*MAT a ade2-1 canl-100 his3-11,15 leu2-3,112 trpl-1 ura3-1 trk1::HIS3 trk2::TRP1*
BYT12	BY4741	*MATa his3* *Δ1 leu2* *Δ0 met15* *Δ0 ura3* *Δ0 trk1* *Δ::loxP trk2* *Δ::loxP*
PLY240	JRY379	*MAT a his3-Δ200 leu2-3,112 trp1-Δ901 ura3-52 suc2-Δ9 trk1Δ51 trk2Δ50::kanMX*
PLY246	JRY379	*MAT a his3-Δ200 leu2-3,112 trp1-Δ901 ura3-52 suc2-Δ9 trk1Δ51 trk2Δ50::kanMX tok1Δ1::HIS3*

**Table 2 ijms-20-02133-t002:** Techniques for detecting protein–protein interaction in yeast (* Cellular compartment where the interaction takes place).

Possible Baits		Year	Technique	Response	Cellular Compartment *	References
Proteins capable of entering nucleus	Non-transactivating proteins	1989	Classic Y2H system (Y2H)	Transcriptional activation	Nucleus	[127]
		1996	Reverse Y2H system (rY2H)	Transcriptional activation	Nucleus	[128,129]
		1996	Yeast three-hybrid system (Y3H)	Transcriptional activation	Nucleus	[130]
		1999	Dual-bait system	Transcriptional activation	Nucleus	[131]
	Transactivating proteins	2001	Repressed transactivator system (RTA)	Inhibition of transcriptional activation	Nucleus	[132]
		2001	RNA polymerase III system (Pol III)	Transcriptional activation	Nucleus	[133]
Membrane proteins		1998	Membrane split-ubiquitin system (MbY2H)	Transcriptional activation	Membrane periphery	[134,135,136]
		2000	Heterotrimeric G-protein fusion system	Inhibition of protein G signaling	Membrane periphery	[137]
		2001	Reverse Ras recruitment system (rRRS)	Ras signaling	Membrane periphery	[138]
Cytosolic proteins		1997	SOS recruitment system (SRS)	Ras signaling	Membrane periphery	[139]
		1998	Ras recruitment system (RRS)	Ras signaling	Membrane periphery	[140]
		2007	Cytosolic split-ubiquitin system (CytoY2H)	Transcriptional activation	Endoplasmic reticulum membrane periphery	[134,141]
Extracellular and secretory pathway proteins		1997	Yeast surface system (YS2H)		Extracellular surface	[142]
		2003	SCINEX-P system	Downstream signaling & transcriptional activation	Endoplasmic reticulum	[143]
		2010	Golgi Y2H system (GY2H)	Och1 activity	Golgi lumen	[144]
Nuclear, membrane and cytosolic proteins		1994	Generally applicable split-ubiquitin system	Uracil auxotrophy and 5-FOA resistance	Cytosol	[134,145]
		1998	Split-mDHFR system	DHFR activity	Native compartment	[146]
		2001	Split-luciferase system	Luminescent signal	Native compartment	[147]
		2004	Split-Trp system	Trp1 activity	Cytosol; Native compartment	[148]
		2005	Split-FP system	Fluorescent signal	Native compartment	[149,150]

**Table 3 ijms-20-02133-t003:** Interactors of K^+^/Na^+^ transporters/channels detected using protein–protein interaction techniques in yeast.

Na^+^/K^+^ Transporter/Channel	Interactors	Technique	References
AKT1	KAT1, AtKC1	MbY2H, mating-based	[156]
	KDC1	MbY2H	[160]
AKT2	MRH1/MDIS2	MbY2H, mating-based	[161]
	SLAC1	MbY2H, mating-based	[162]
OsHKT1	OsCNIH1	MbY2H, mating-based	[163]
KAT1	KAT1, AKT1, PUP11	MbY2H, mating-based	[156]
	SLAC1	MbY2H, mating-based	[162]
	VAMP721	MbY2H, mating-based	[164]
KAT2	SLAC1	MbY2H, mating-based	[162]
AtKC1	AKT1, NRT2.7, ROP1	MbY2H, mating-based	[156]
	SLAC1	MbY2H, mating-based	[162]
	SYP121	MbY2H, mating-based	[165,166]
	VAMP721	MbY2H, mating-based	[164]
KDC1	AKT1	MbY2H	[160]
KUP6	SnRK2.6 , SnRK2.2	MbY2H, mating-based	[167]
AKT1	AIP1, CIPK6, CIPK16	Y2H	[168]
	AKT1, AKT2, AtKC1	Y2H	[169]
	CBL10 (CBL5, CBL7)	Y2H	[170]
	CIPK23	Y2H	[171,172,173]
		Y2H competition assay	[170]
AKT1, OsAKT1, PutAKT1	KPutB1, OsKOB1	Y2H	[174]
AKT2	AKT1, AKT2, AtKC1	Y2H	[169]
	CIPK6	Y2H	[175]
	MRH1/MDIS2	Y2H Matchmaker Gold	[161]
	PP2CA	Y2H	[176]
AKT3	AtPP2CA	Y2H	[177]
GORK	AtPP2CA	Y2H	[178]
	GORK, SKOR	Y2H	[179]
OsHAK1	OsRUPO	Y2H	[180]
KAT1	KDC1	Y2H	[181]
VvKAT1	VvSnRK2.4	Y2H	[182]
OsKAT2	OsKAT2, OsKAT3	Y2H	[183]
OsKAT3	OsKAT2, OsKAT3	Y2H	[183]
AtKC1	AKT1	Y2H	[169]
KDC1	KAT1	Y2H	[181]
OsKOB1	AKT1, OsAKT1, PutAKT1	Y2H	[174]
KPutB1	AKT1, OsAKT1, PutAKT1	Y2H	[174]
KST1	SKT2, SKT3	Y2H	[184]
SKOR	SKOR, GORK	Y2H	[179]
SKT2	KST1	Y2H	[184]
SKT3	KST1	Y2H	[184]
TRH1	TRH1	Y2H	[185]

**Table 4 ijms-20-02133-t004:** Oligomers of K^+^/Na^+^ transporters/channels detected using protein–protein interaction techniques in yeast.

Oligomer		References
AKT1	AKT1, AKT2	[169]
	KAT1	[156]
	AtKC1	[156,169]
	KDC1	[160]
AKT1, OsAKT1, PutAKT1	KPutB1, OsKOB1	[174]
AKT2	AKT1, AKT2	[169]
	AtKC1	[169]
GORK	GORK, SKOR	[179]
KAT1	AKT1	[156]
	KAT1	[156]
	KDC1	[181]
OsKAT2	OsKAT2, OsKAT3	[183]
OsKAT3	OsKAT2, OsKAT3	[183]
AtKC1	AKT1	[156,169]
KDC1	AKT1	[160]
	KAT1	[181]
OsKOB1	AKT1, OsAKT1, PutAKT1	[174]
KPutB1	AKT1, OsAKT1, PutAKT1	[174]
KST1	SKT2, SKT3	[184]
SKOR	SKOR, GORK	[179]
SKT2	KST1	[184]
SKT3	KST1	[184]
TRH1	TRH1	[185]

**Table 5 ijms-20-02133-t005:** Regulatory proteins of K^+^/Na^+^ transporters/channels detected using protein–protein interaction techniques in yeast. Light gray, kinases and phosphatases; medium gray, channels and transporters; dark gray targeting-related proteins.

Na^+^/K^+^ Transporter/Channel	Regulatory Protein	Na^+^/K^+^ Transporter/Channel Regulation	References
AKT1	AIP1	Reduces AKT1 activity	[168]
	CBL10 (CBL5, CBL7)	Impairs AKT1 activity	[170]
	CIPK6, CIPK16	Phosphorylates and activates AKT1	[168]
	CIPK23	Phosphorylates and activates AKT1	[170,171,172,173]
AKT2	CIPK6	Upon interaction with CIPK6, CBL4 mediates ER-to-PM translocation of AKT2 and enhances AKT2 activity	[175]
	MRH1/MDIS2		[161]
	PP2CA	Dephosphorylates and inhibits AKT2, regulated by ABA signaling	[176]
	SLAC1		[162]
AKT3	AtPP2CA		[177]
GORK	AtPP2CA	Dephosphorylation-independent inactivation of GORK	[178]
OsHAK1	OsRUPO	Disruption of RUPO leads to K^+^ over-accumulation in pollen	[180]
OsHKT1	OsCNIH1	Golgi-localization of OsHKT1	[163]
KAT1	PUP11		[156]
	SLAC1	Inhibits KAT1 activity	[162]
	VAMP721	Suppresses KAT1 and KC1 activity	[164]
VvKAT1	VvSnRK2.4		[182]
KAT2	SLAC1		[162]
AtKC1	NRT2.7	K^+^ is known to increase nitrate (NO_3_^−^ ) uptake from soil	[156]
	ROP1	Actin filament reorganization affects K^+^ channel activities in stomata. ROP1 regulates pollen tip growth	[156]
	SLAC1		[162]
	SYP121	Promotes KAT1 activity, in the presence of KC1	[165,166]
	VAMP721	Suppresses KAT1 and KC1 activity	[164]
KUP6	SnRK2.6, SnRK2.2	SnRK2.6 phosphorylates KUP6, regulated by ABA signaling (drought stress)	[167]

**Table 6 ijms-20-02133-t006:** SOS genes from various plant species characterized in yeast.

Plant Species	Genes Characterized	Reference
*Arabidopsis thaliana*	SOS1-3	[197]
*Chrysanthemum crassum*	SOS1	[206]
*Cymodocea nodosa*	SOS1	[207]
*Eutrema salsugineum*	SOS1	[208]
*Glycine max*	SOS1	[209]
*Oryza sativa*	SOS1-3	[210]
*Phragmites australis Trinius*	SOS1	[211]
*Physcomitrella patens*	SOS1	[212]
*Populus trichocarpa*	SOS1-3	[213]
*Schrenkiella parvula*	SOS1	[208]
*Sesuvium portulacastrum*	SOS1	[214]
*Solanum lycopersicum*	SOS2	[215]
*Triticum aestivum*	SOS1	[216]
*Triticum durum*	SOS1	[217]

**Table 7 ijms-20-02133-t007:** A summary of the heterologous overexpression screenings performed in yeast using plant cDNA libraries.

Plant Species	cDNA Library [Reference]	Yeast Strain	Screening Conditions	Isolated Genes	Reference
*Arabidopsis thaliana*	Leaves from Arabidopsis seedlings [239]	*ena1-4* (W303-1A)	50 mM LiCl	AtRCY1 (K/T-cyclin with SR domain of splicing proteins) AtSRL1 (SR domain of splicing proteins)	[237]
				AtLTL1 (GDSL-motif lipase)	[240]
	[241]	*cna1cna2*YPH499	200 mM NaCl	AtSTZ1 (Zinc finger protein) AtSTO1 (B-Box domain protein 24)	[242]
*Atriplex canescens*	Young leaves and stems treated with 400 mM NaCl [243]	WT (INVSc1)	2 M NaCl	KJ026992 (Cyclophilin) KJ027014 (Glycine-rich protein) KJ027023 (Cytochrome P450) KJ027035 (Temperature-induced lipocalin) KJ027049 (Cysteine proteinase A494) KJ027057 (Alanine aminotransferase 2) KJ027061 (Hexose transporter) KJ027088 (RNA-binding family protein) KJ027102 (Cysteine proteinases) KJ027110 (calmodulin1)	[244]
*Beta vulgaris*	Leafs from salt stressed plants [245]	*ena1-4 nha1*(W303-1A)	150 mM NaCl	BvCK2 (catalytic subunit of the casein kinase)	[245]
				BveIF1A (Translation initiation factor)	[246]
				BvSATO1 (RNA binding protein with RGG and RE/D motifs) BvSATO2 (homologous to SATO1) BvSATO4 (RNA binding protein) BvSATO5 (RNA binding protein) BvU2AF (U2snRNP AF protein)	[247]
		*gpd1*(W303-1A)	1 M Sorbitol	BvSAT1 (Serine acetyl trasferase 1)	[248]
				BvGLB2 (Type II non symbiotic plant hemoglobin)	[249]
		WT (W303-1A)	10 °C	BvCOLD1 (TIP-like aquaporin)	[250]
*Ipomoea pes-caprae*	Growing leaves, shoots and roots [251]	*ena1-4 nha1 nhx1*(W303-1A)	75 mM NaCl	MF680587 (putative abscisic acid, stress, and ripening-induced protein (ASR)) MF680589 (nudix hydrolase, chloroplastic) MF680592 (F-box protein At5g46170-like) MF680597 (fructokinase) MF680602 (adenosylhomocysteinase 1) MF680603 (peptide upstream protein) MF680604 (catalase) MF680608 (40S ribosomal protein S25) MF680611 (phosphomannomutase) MF765747 (dnaj protein-like protein) MF680614 (phytosulfokines-like) MF680616 (sugar carrier protein C) MF680619 (abscisic acid 8′-hydroxylase 4) KX426069 (dehydrin)	[251]
*Jatropha curcas*	3-4 week seedlings treated with 150 mM NaCl [252]	*shs-2 *(UV-generated mutant in the BY4741 background)	750 mM NaCl	FJ489601 (Allene oxide cyclase) FJ489602 (Thioredoxin H-type (TRX-h)) FJ489603 (Metallothionein) FJ489604 (Heterotrophic ferredoxin) FJ489605 (Defensin) FJ489606 (Calmodulin-7 (CAM-7)) FJ489608 (S18.A ribosomal protein) FJ489609 (60S ribosomal protein L18a) FJ489611 (Unknown protein) FJ619041 (Membrane protein -2) FJ619045 (Profilin-like protein) FJ619048 (Copper chaperone) FJ619052 (Annexin-like protein) FJ619053 (Al-induced protein) FJ619055 (60S ribosomal protein L39) FJ619056 (Ribosomal protein L37) FJ619057 (Ribosomal protein L15) FJ623457 (40S ribosomal protein S15) FJ623458 (40S ribosomal S18)	[252]
*Oryza sativa*	Leaves from seedlings treated with different abiotic stresses [253]	WT (AH109)	900 mM NaCl	OsMPG1 (mannose-1-phosphate guanyl transferase gene)	[253]
*Paspalum vaginatum*	Cultivated stolons treated with 250 mM NaCl [254]	*ena1-4*(G19)	500 mM NaCl	KT203435 (Uncharacterized protein) KT203436 (Iron-regulated transporter) KT203439 (Early light-induced protein) KT203440 (14-3-3-Like protein) KT203441 (Class 1 HSP) KT203442 (Cysteine synthase) KT203443 (Aldo-ketoreductase) KT203444 (L-Ascorbate peroxidase 2) KT203447 (Nop14-like family protein) KT203450 (Protein IQ-DOMAIN 14-like) KT203451 (Metacaspase-5-like)	[254]
*Phoenix datilifera*	Salt-treated roots [255]	WT (INvSc1)	2 M NaCl	XM_008806660.2 (11S globulin seed storage protein 2-like) XM_008805834.2 (ABC transporter) XM_008780694.2 (Aquaporin PIP1-2) XM_008793314.1 (Aquaporin PIP2-4-like) XM_008779330.1 (Cysteine desulfurase) XM_008802947.2 (Cytochrome b5-like) XM_008783561.2 (Cytospin-A-like 1) XM_008797620.2 (Hexokinase-2-like) XM_008780092.2 (Mavicyanin-like 1) XM_008814440.2 (Peroxidase 3-like) XM_008786338.2 (Peroxidase 3-like 2) XR_604439.2 (Uncharacterized) XM_008800513.2 (Uncharacterized) XM_017846106.1 (Uncharacterized) XM_017846788.1 (Uncharacterized)	[255]
*Salicornia europaea*	3-month-old plants	WT (BY4741)	1.6 M NaCl	SeNN8 (Similar to FKBP5) SeNN24 (Thaumatin like protein) SeNN43 (Unknown protein)	[256]
*Solanum tuberosum*	Plants grown in vitro and subjected to heat shock at 35 °C [257]	WT (BY4741)	39 °C	StnsLTP1 (Non-specific Lipid Transfer Protein-1)	[258]
*Zea mays*	Maize kernels	Not indicated	Not indicated	MBF1a (Multiprotein bridging factor 1a transcriptional coactivator)	[259]
*Zoysia matrella*	Cultivated stolons treated with 300 mM NaCl	*ena1-4* (G19)	500 mM NaCl	KM265171 (Uncharacterized protein) KM265174 (C2H2-type Zinc finger protein) KM265176 (Unknown protein) KM265177 (Alcohol dehydrogenase 1) KM265179 (Protein disulfide isomerase) KM265182 (Glyoxylate reductase) KM265183 (Serine carboxypeptidase)	[260]

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
