# Peer review of "Saccharomyces cerevisiae as a Tool to Investigate Plant Potassium and Sodium Transporters"

_ijms, 2019, doi:10.3390/ijms20092133_

Round 1
Reviewer 1 Report
The authors review multiple cases of K+ and Na+ channels and transporters expressed in the plasma membrane of S. Cerevisiae. The motivation is stated to be understanding Na+/K+ homeostasis. However, given that the system is modified by non-native channels, hence will not behave like a normal system establishing homeostasis, what they really are reviewing is the behavior of the channels and transporters that can be reconstituted in S. Cerevisiae. This said, they have provided what appears to be a rather thorough and comprehensive review of the cases in which this reconstituted system has been studied. This is likely to be quite useful to workers in the field, and so should be published, even though it does not really address its stated aim. The paper would be improved if the authors made clear that they were reviewing the use of S. Cerevisiae as a system for expressing the membrane proteins and studying their function, rather than reviewing homeostasis. It is fair to state that the mechanism by which homeostasis is achieved requires these proteins, and understanding their function is needed to understand homeostasis; this is (sort of) implied, but it would be better to make it explicit. The title of the paper instead implies that the paper considers homeostasis directly.
The writing is not a problem.
I did not proofread the paper, but did notice a couple of typos anyway.
Line 21) missing “of” (…of genes..)
Line 589) “identified” should be “identify”
Line 674) missing “it” (…that it is…)
If I noticed these typos without looking for any, there are probably many more; the paper should be proofread.
Author Response
RESPONSES TO REVIEWERS:
Reviewer 1:
The authors review multiple cases of K+ and Na+ channels and transporters expressed in the plasma membrane of S. Cerevisiae. The motivation is stated to be understanding Na+/K+ homeostasis. However, given that the system is modified by non-native channels, hence will not behave like a normal system establishing homeostasis, what they really are reviewing is the behavior of the channels and transporters that can be reconstituted in S. Cerevisiae. This said, they have provided what appears to be a rather thorough and comprehensive review of the cases in which this reconstituted system has been studied. This is likely to be quite useful to workers in the field, and so should be published, even though it does not really address its stated aim. The paper would be improved if the authors made clear that they were reviewing the use of S. Cerevisiae as a system for expressing the membrane proteins and studying their function, rather than reviewing homeostasis. It is fair to state that the mechanism by which homeostasis is achieved requires these proteins, and understanding their function is needed to understand homeostasis; this is (sort of) implied, but it would be better to make it explicit. The title of the paper instead implies that the paper considers homeostasis directly.
The writing is not a problem.
I did not proofread the paper, but did notice a couple of typos anyway.
Line 21) missing “of” (…of genes..)
Line 589) “identified” should be “identify”
Line 674) missing “it” (…that it is…)
If I noticed these typos without looking for any, there are probably many more; the paper should be proofread.
Responses: We thank the reviewer for his/her helpful comments. We agree with their assessment and have changed the title accordingly to reflect this.
We have also corrected the typos pointed out by the reviewer and have thoroughly proof-read the manuscript and corrected all additional errors encountered.
Reviewer 2 Report
I have read your interesting, well planned and quite comprehensive description of the use of a model system Saccharomyces cerevisiae for studies of sodium and potassium homeostasis in plants. Although I have found no major deficiencies, I can offer some suggestions that in my opinion will clarify and improve the impact of this work when it is published. In addition, I have listed some editorial changes that are needed.
In terms of general points I think that you should consider adding the word 'transport' to your title and emphasizing this in a number of different places in the manuscript. In almost all sections involving ion channel or antiporter expression the functional protein has important transport roles and therefore contributes to homeostasis. A second general point that would involve some additional work would be to add papers that set out the known roles of changes in intracellular calcium in plant function and presumable in the model system of your choice. One example where this is important is in the sections of the manuscript dealing with modulation of channel function by phosphatase activity. In most mammalian systems a number of these key phosphatases are also calcium-dependent. Recent publications that draw attention to the kind of changes that I think should be referred to are:
Hander T....Stael S. Damage on plants activates CA2+-dependent metacaspases for release of immunomodulatory peptides. Science 2019, 363: 1301.
Toyota M...Gilroy S. Glutamate triggers long-distance, calcium-based plant defines signalling. Science 2018, 361:1112-1116.
Another general point and recommendation would be to include a section concerning pH regulation in plants and in your model system. Two reasons for this is that you refer quite extensively to the interesting and important functions of inwardly rectifying potassium channels and sodium hydrogen antiporters but do not put the identification of these families of integral membrane proteins in context based on their known pH dependence.
Finally, I would guess that a number of the ion channels and antiporters that you have selected for emphasis are modified by intracellular phosphoinositide metabolism. If so, the reader would be interested in knowing whether and to what extent your model system and your chosen examples from plants replicate or mimic what is known for PIP2 and PIP4 modulation of mammalian ion channels and pumps.
The remainder of my comments are suggestions for wording changes.
Line 14 - The word 'intracellular' should be added before your word 'high'.
Line 22 - Your word 'on' should be replaced by 'in'.
Line 33 - The term 'macronutrient' should be defined as it applies to plant physiology.
Line 38 - Your word 'of' should be replaced with 'across'.
Line 87 - Your word 'revisions' should be replaced with 'reviews'.
Line 105 - Your word 'extensive' should be replaced with 'comprehensive'.
Line 138 - My understanding is that this Figure includes both antiporters and ion channels.
Lines 562-564 - It is with respect to this Table and this section of your text that my suggestion for information concerning regulation of intracellular calcium and the calcium-dependence of enzyme activity is perhaps most relevant.
Line 783 - There should be a space after the word 'modulation'.
Line 810 - References 33 and 34 are incomplete.
Line 857 - Reference 55 is incomplete.
Line 872 and line 911 - These references are incomplete.
Line 973 - I was intrigued by this reference. Does this paper establish the presence of an ortholog to or variant of one of the mammalian sodium potassium ATPases? If so, a short section should be added to your review.
Line 1198 - This reference and related text provide one of the reasons for me suggesting that additional information should be added concerning sodium and/or potassium-dependent regulation of intracellular calcium in plants.
Author Response
RESPONSES TO REVIEWERS:
Reviewer 2:
I have read your interesting, well planned and quite comprehensive description of the use of a model system Saccharomyces cerevisiae for studies of sodium and potassium homeostasis in plants. Although I have found no major deficiencies, I can offer some suggestions that in my opinion will clarify and improve the impact of this work when it is published. In addition, I have listed some editorial changes that are needed.
In terms of general points I think that you should consider adding the word 'transport' to your title and emphasizing this in a number of different places in the manuscript. In almost all sections involving ion channel or antiporter expression the functional protein has important transport roles and therefore contributes to homeostasis.
Response: We thank the reviewer for this helpful comment. We have made the corresponding changes.
A second general point that would involve some additional work would be to add papers that set out the known roles of changes in intracellular calcium in plant function and presumable in the model system of your choice. One example where this is important is in the sections of the manuscript dealing with modulation of channel function by phosphatase activity. In most mammalian systems a number of these key phosphatases are also calcium-dependent. Recent publications that draw attention to the kind of changes that I think should be referred to are:
Hander T....Stael S. Damage on plants activates CA2+-dependent metacaspases for release of immunomodulatory peptides. Science 2019, 363: 1301.
Toyota M...Gilroy S. Glutamate triggers long-distance, calcium-based plant defines signalling. Science 2018, 361:1112-1116.
Response: This is an interesting point. We have included a paragraph addressing selected aspects of the calcium-dependent regulation of potassium channels (lines 593-599).
Another general point and recommendation would be to include a section concerning pH regulation in plants and in your model system. Two reasons for this is that you refer quite extensively to the interesting and important functions of inwardly rectifying potassium channels and sodium hydrogen antiporters but do not put the identification of these families of integral membrane proteins in context based on their known pH dependence.
Response: Although we consider a full discussion of pH regulation in plants to be beyond the scope of the current review, we have included relevant information regarding pH and the analysis of the plant proton ATPases in yeast (Lines 656-672).
Finally, I would guess that a number of the ion channels and antiporters that you have selected for emphasis are modified by intracellular phosphoinositide metabolism. If so, the reader would be interested in knowing whether and to what extent your model system and your chosen examples from plants replicate or mimic what is known for PIP2 and PIP4 modulation of mammalian ion channels and pumps.
Response: This is a very interesting question indeed. We have revised the literature and have not found any studies addressing the role of PIP2 or PIP4 in plant potassium channel regulation in yeast. There is evidence of PIPs regulating several processes in yeast, including actin cytoskeleton and intracellular trafficking (see Odorizzi, et al., TIBS 2000; Stahl and Thorner BBA 2007, for example), but we found no reports addressing their role in the direct regulation of ion transporters in yeast or for plant proteins heterologously expressed in yeast. It would be an extremely interesting line of investigation for the future.
The remainder of my comments are suggestions for wording changes.
Line 14 - The word 'intracellular' should be added before your word 'high'.
Line 22 - Your word 'on' should be replaced by 'in'.
Line 33 - The term 'macronutrient' should be defined as it applies to plant physiology.
Line 38 - Your word 'of' should be replaced with 'across'.
Line 87 - Your word 'revisions' should be replaced with 'reviews'.
Line 105 - Your word 'extensive' should be replaced with 'comprehensive'.
Line 138 - My understanding is that this Figure includes both antiporters and ion channels.
Lines 562-564 - It is with respect to this Table and this section of your text that my suggestion for information concerning regulation of intracellular calcium and the calcium-dependence of enzyme activity is perhaps most relevant.
Line 783 - There should be a space after the word 'modulation'.
Line 810 - References 33 and 34 are incomplete.
Line 857 - Reference 55 is incomplete.
Line 872 and line 911 - These references are incomplete.
Response: We have made the corresponding changes.
Line 973 - I was intrigued by this reference. Does this paper establish the presence of an ortholog to or variant of one of the mammalian sodium potassium ATPases? If so, a short section should be added to your review.
Response: This reference refers to a Na+-K+/H+ antiporter, not a sodium potassium ATPase.
Line 1198 - This reference and related text provide one of the reasons for me suggesting that additional information should be added concerning sodium and/or potassium-dependent regulation of intracellular calcium in plants.
Response: See answer to the related question above.